# *Eurotium cristatum* from Fu Brick Tea Promotes Adipose Thermogenesis by Boosting Colonic *Akkermansia muciniphila* in High-Fat-Fed Obese Mice

**DOI:** 10.3390/foods12203716

**Published:** 2023-10-10

**Authors:** Yu Wang, Ting Li, Chengcheng Yang, Yingmei Wu, Yueyue Liu, Xingbin Yang

**Affiliations:** College of Food Engineering and Nutritional Science, Shaanxi Normal University, Xi’an 710119, Chinatingli@snnu.edu.cn (T.L.);

**Keywords:** Fu brick tea, *Eurotium cristatum*, fat thermogenesis, gut microbiota, *Akkermansia muciniphila*

## Abstract

This study investigated the potential fat-thermogenic effects of *Eurotium cristatum*, and elucidated the underlying mechanisms. The 12-week administration of *E. cristatum* in HFD-fed obese mice reduced body weight and improved glucolipid metabolism disorders. The administration of *E. cristatum* also efficiently promoted thermogenesis by increasing the expression of UCP1 and PRDM16 in both interscapular brown adipose tissue (iBAT) and inguinal white adipose tissue (iWAT) of HFD-fed mice. Furthermore, *E. cristatum* shaped the gut microbiome by increasing the abundance of *Parabacteroides* and *Akkermansia muciniphila*, and also elevated the levels of cecal short-chain fatty acids, particularly propionate and acetate. Of note, *A. muciniphila* was highly negatively correlated with body weight gain (r = −0.801, *p* < 0.05) and the iWAT index (r = −0.977, *p* < 0.01), suggesting that *A. muciniphila* may play an important role in the thermogenic mobilization induced by *E. cristatum*. Continuous supplementation with *A. muciniphila* suppressed adipose accumulation, improved glucolipid metabolism, and enhanced the thermogenic activity of iWAT and iBAT. Collectively, our results propose that boosted *A. muciniphila* acts as a key microbe in tea-derived probiotic *E. cristatum*-mediated fat-thermogenic and anti-obesity effects.

## 1. Introduction

Fu brick tea (FBT) is a distinctive type of fermented dark tea that undergoes a complex manufacturing process involving multiple stages, including steaming, piling, pressing, and fermentation (microbial growth) [1]. Microbial fermentation plays a critical role in the development of the unique flavor and quality characteristics of FBT. During this process, *E. cristatum* is the dominant fungus responsible for creating the “golden flower” bloom on the surface of FBT following microbial community succession [2]. Therefore, the amount of *E. cristatum* during fermentation serves as an important indicator for evaluating the quality of FBT. *E. cristatum* not only absorbs nutrients from tea leaves to promote its growth during the fermentation process but also triggers a series of reactions that contribute to the development of FBT’s distinct color, aroma, and flavor [3]. For instance, *E. cristatum* secretes polyphenol oxidases, which facilitate the oxidation of catechins and their gallic acid derivatives present in FBT. This enzymatic action leads to the generation of tea pigments such as theaflavins (TFs), thearubigins (TRs), and theabrownins (TBs). TFs undergo further oxidation to TRs, which then polymerize to form TBs [4,5]. The formation of TBs continually increases during this process and exhibits high synchronicity with the growth of *E. cristatum* [5]. The role of FBT in reducing obesity by regulating the gut microbiota is increasingly recognized, and our previous study has shown that TBs as tea pigments derived from FBT can enhance adipocyte thermogenic and anti-inflammatory effects in obese mice by modulating SCFAs [6]. Additionally, preliminary studies have shown that *E. cristatum* can significantly reduce fat deposition and exert hypolipidemic activity in vitro and in vivo [7,8]. From these perspectives, it is hypothesized that *E. cristatum* plays a significant regulatory role in the synthesis of TBs, and may serve as a crucial nutritional substance that can induce adipose thermogenic activity, potentially aiding in the prevention of obesity.

Adipose tissue is responsible for regulating metabolism and homeostasis by secreting various bioactive adipokines [9,10]. The pathogenesis of obesity mainly stems from an energy imbalance, which leads to excess fat storage [9,10,11]. Consequently, different types of adipose tissue can re-establish the energy balance to counter obesity via the storage and release of lipids [12]. White adipose tissue (WAT) with unilocular lipid droplets is responsible for storing energy and maintaining the relative balance of human energy, and high-fat diet (HFD)-induced obese individuals specialize with large deposits of WAT [11,13,14]. Unlike WAT, brown adipose tissue (BAT) is an energy-consuming adipocyte tissue with multilocular lipid droplets and many mitochondria [12,15]. Thermogenic brown adipose tissue (BAT) is distinguished by the expression of uncoupling protein 1 (UCP1), which is a proton transporter that dissipates energy as heat via uncoupling oxidative phosphorylation [15,16]. Interestingly, another type of thermogenic adipose tissue found in WAT is termed beige adipocytes. This “brown-like” adipocyte exerts similar characteristics to BAT by upregulating the expression of UCP1, PR domain-containing 16 (PRDM16), and peroxisome proliferator-activated receptor-γ coactivator 1-α (PGC1-α) [17,18]. Previous studies have shown that enhancing BAT activity and converting WAT into beige adipocytes can provide novel approaches for preventing obesity [19,20,21].

Retrospective studies commonly report that the gut microbiota and its metabolites, particularly short-chain fatty acids (SCFAs), promote adipose thermogenesis as a mechanism for counteracting obesity and related metabolic disorders [22,23]. Diet plays a significant role in shaping the composition of the gut microbiota, and the interaction of the gut microbiota with diet can affect the host’s physiological function and nutrient metabolism [23,24]. Previous studies showed that HFD-induced obese subjects exhibited a less diverse gut microbiota than normal diet-based individuals, and HFD lowered the relative abundance of beneficial bacteria involved in regulating energy balance [25,26]. For example, Noni fruit polysaccharide (NFP) ameliorated HFD-induced obesity by increasing the formation of bile acids and remodeling the gut microbiota, in which the relative abundances of the phylum *Bacteroidetes* and genera *Lactobacillus*, *Bilophila*, and *A. muciniphila* increased after administration of NFP [26]. *A. muciniphila* has the ability to improve metabolism, and supplementation with *A. muciniphila* can activate the secretion of the gastrointestinal hormone glucagon-like peptide 1, a gastrointestinal hormone that plays a vital role in improving glucose homeostasis in mice [27]. Emerging evidence has indicated that certain stimulants reinforce the effect of thermogenesis on dietary obesity by increasing the relative abundances of beneficial bacterial taxa such as *Butyricicoccus*, *Lactobacillus*, and *A. muciniphila*, suggesting that gut microbiota, including *A. muciniphila*, are important factors affecting the process of fat thermogenesis [27,28]. However, it is still unclear which specific beneficial bacterium might be able to influence the gut microbial ecosystem and thermogenic function of adipocytes.

Therefore, the purpose of the present study was to investigate the impact of *E. cristatum* on the thermogenic effects of adipose tissues, including interscapular brown adipose tissue (iBAT) and inguinal white adipose tissue (iWAT), as well as the underlying mechanism of *E. cristatum* in regulating adipose thermogenesis mediated by the gut microbiota, with special emphasis on the critical species *A. muciniphila*. 

## 2. Materials and Methods

### 2.1. Chemicals and Reagents

*E. cristatum* obtained from the China Center of Industrial Culture Collection (Beijing, China) was cultured according to our previous method [8]. The stored *E. cristatum* was activated and cultured on potato dextrose agar (PDA) medium at 28 °C for 72 h. The harvested spores were mixed with distilled water and counted using a hemocytometer. Based on the counting concentration, the working concentration was adjusted to 10^6^ and 10^8^ CFU/mL using distilled water. For the colon-derived *A. muciniphila,* it was grown in a brain–heart infusion medium under strict anaerobic conditions using a suitable anaerobic chamber (100% N_2_). The culture was maintained at 37 °C for 48 h, followed by storage at −80 °C for future use. The BCA Protein Assay Kits, total cholesterol (TC), total triglycerides (TG), high-density lipoprotein cholesterol (HDL-C), and low-density lipoprotein cholesterol (LDL-C) were purchased from Jiancheng Bioengineering Institute Co. Ltd. (Nanjing, China). The antibodies UCP1 (ab10983, 1:500 dilution), PRDM16 (ab106410, 1:500 dilution), and Cy3-conjugated IgG (ab97075, 1:200 dilution) were purchased from Abcam (Cambridge, MA, USA). Sangon Biotech (Shanghai, China) provided the TRIzol reagent, reverse transcription kit, and SYBR green mixture.

### 2.2. Animal Experiments 

Healthy male C57BL/6J mice (4 weeks old) were purchased from the Experimental Animal Center of Air Force Medical University (Xi’an, China). All animal experiments were conducted following the ethical requirements of the Committee on Care and Use of Laboratory Animal of Shaanxi Normal University (C31671823, Xi’an, China). All mice were housed in a controlled environment with a temperature maintained at 22 ± 2 °C and a humidity level between 55 and 65%. They followed a 12 h light/dark cycle. During the entire experiment, the mice had unrestricted access to both water and food. The amount of food consumed and the body weights of the mice were recorded on a weekly basis. In the present work, two independent experiments were conducted: (1) Assessing the preventive effects of *E. cristatum* on HFD-induced obesity. C57BL/6J mice (*n* = 40) were randomly assigned to four groups (*n* = 10) after 1 week of acclimation period: the normal diet (ND) group, high-fat diet (HFD) group, HFD plus 10^6^ CFU/mL *E. cristatum* (HFD-ECL) group, HFD plus 10^8^ CFU/mL *E. cristatum* (HFD-ECH) group. (2) Investigating the role of gut *A. muciniphila* in the preventive effects of *E. cristatum* against obesity. C57BL/6J mice (*n* = 30) were randomly assigned to three groups (*n* = 10) after 1 week of acclimation period: the normal diet (ND) group, high-fat diet (HFD) group, HFD plus 10^8^ CFU/mL *A. muciniphila* (HFD-AKK) group. 

The HFD, HFD-ECL, HFD-ECH, HFD, and HFD-AKK groups were fed a high-fat diet containing 30% lard, whereas the ND group received a normal diet containing 0% lard [29]. The mice of the ND and HFD groups were orally administered 0.2 mL distilled water once daily for 12 weeks, and the mice of the HFD-ECL and HFD-ECH groups were orally administered 0.2 mL *E. cristatum* once daily for 12 weeks, while the mice of the HFD-AKK group were orally administered 0.2 mL *A. muciniphila* once daily for 12 weeks. 

### 2.3. Oral Glucose Tolerance Test (OGTT) and Insulin Tolerance Test (ITT)

Blood samples were taken from a tail clip after a 6 h fasting period. Measurements at 15, 30, 60, and 90 min after either the intraperitoneal injection of insulin (0.75 unit/kg bw) or the administration of glucose (2.0 g/kg bw) were used to test the fasting blood glucose levels.

### 2.4. Analysis of Serum Biochemical Assessment

After allowing the blood samples to stand at room temperature for 1 h and centrifuging them at 1000× *g* for 15 min, the samples were mixed with the appropriate reagents according to the instructions provided by the manufacturer. The mixtures were thoroughly mixed and incubated for the designated periods at controlled temperatures. After the incubation period, the absorbance of the reaction mixtures was measured at specific wavelengths using a spectrophotometer. The concentrations of serum biochemical indices were determined using calibration curves provided by the experimental operation kits. 

### 2.5. Examination of Histopathology and Immunofluorescence 

Fresh standardized adipose tissues were submerged in 10% formalin and then embedded in paraffin. Hematoxylin–eosin (H&E) staining was used to visualize tissue morphology in thin 5 µm sections taken from the embedded tissues. Immunofluorescence staining was subsequently performed following the previous study, with the primary antibodies (UCP1, PRM16) diluted proportionally as necessary [30]. The tissue sections underwent a series of processes including dewaxing, rehydration, and overnight incubation at 4 °C with the primary antibodies. Following this, the sections were washed and incubated with a secondary antibody that was labeled with Cy3-conjugated immunoglobulin. Additionally, the nuclei were counterstained with DAPI to visualize cell nuclei. Immunohistochemical images of the stained tissues were then observed using an Axio Imager Upright Microscope (Axio Imager M2, ZEISS, Oberkochen, Germany).

### 2.6. qRT-PCR Analysis

The TRIzol reagent was employed to extract total RNA. Following this, double-stranded cDNA synthesis was carried out as per the instructions provided with the reverse transcription kit. The resulting cDNA served as a template for PCR amplification. To quantify the target sequence, a SYBR green mix was utilized. The primer sequences used for amplifying the target genes can be found in Table 1. To analyze the data, β-actin was used as an internal control for normalization, and the 2^−ΔΔCT^ method was employed [31].

### 2.7. Measurement of SCFAs 

The level of SCFAs in the mouse colon was determined using GC-MS (7890A-7000B; Agilent Technologies Inc., Santa Clara, CA, USA) [32]. Briefly, after homogenizing 50 mg of colonic samples with 1 mL of 0.5% phosphoric acid, the supernatant was centrifuged for 10 min at 12,000 rpm to yield 500 μL. In addition, 500 μL of ethyl acetate was added and mixed for 2 min, followed by another centrifugation at 12,000 rpm for 10 min. A flow rate of 1.0 mL/min was used along with a column temperature of 50 °C and an injection volume of 20 μL. Analysis of the samples was performed using a mass detector (QP-2010ULTRA, SHIMADZU, Kyoto, Japan). 

### 2.8. 16S rRNA Sequencing

An E.Z.N.A. Stool DNA Kit was used to extract fecal RNA. Subsequently, the extracted genomic DNA was assessed by subjecting it to 1% agarose gel electrophoresis. The hypervariable V3-V4 region of the 16S rRNA gene was selected for amplicon analysis, and paired-end 300 bp sequencing was performed on a MiSeq platform (Illumina, CA, USA). Amplification was carried out using a forward primer (338F: ACTCCTACGGGAGGCAGCAG) and reverse primer (806R: GGACTACHVGGGTWTCTAAT). Sequencing was conducted on an Illumina Miseq PE 2500 platform. After sequencing, the paired-end reads were split using Illumina sequencing. The double-ended reads were subjected to quality control and filtering based on sequence quality. They were then spliced together based on the overlapping relationship of the paired reads to obtain optimized data after quality control and splicing [30]. The optimized data underwent sequence denoising to obtain representative amplicon sequence variant (ASV) sequences and their corresponding abundance information. These representative ASV sequences and abundance information can be used for various statistical or visual analyses, such as species taxonomic analysis, community diversity analysis, and species difference and correlation analysis. To classify each 16S rRNA gene sequence taxonomically, we employed the RDP Classifier algorithm in conjunction with the Silva (SSU123) 16S rRNA database. During the classification process, a confidence threshold of 70% was utilized to ensure accurate taxonomic assignments. 

### 2.9. Statistical Analysis 

Data were expressed as means ± standard deviation (SD). Statistical analysis was performed using SPSS version 19.0. One-way ANOVA and the Student’s *t*-test were used to determine statistical differences between groups. A *p* value < 0.05 was considered statistically significant. Graphs were generated using GraphPad Prism 8.0. A partial least-squares regression correlation analysis was conducted using R v4.0.4 with the PLS package.

## 3. Results

### 3.1. Effects of E. cristatum on Body Weight and Glucolipid Metabolism in HFD-Fed Obese Mice

There were no significant differences in the initial body weights of mice between the groups, as shown in Figure 1A. Compared to the ND group, the HFD group had a higher body weight growth rate. Remarkably, after intervention for 12 weeks, 10^6^ and 10^8^ CFU/mL of *E. cristatum* resulted in a reduction in body weight of HFD-fed mice from 31.6 ± 1.5 g to 30.9 ± 1.6 g and 29.9 ± 1.7 g, respectively (Figure 1B), despite no significant differences in food intake among the obese mice (Figure 1C).

The OGTT results showed that HFD-fed mice consistently exhibited a higher glycemic response than ND mice, and administration of *E. cristatum* slightly improved impaired glucose tolerance in obese mice (Figure 1D). For the ITT results depicted in Figure 1E, *E. cristatum* administration ameliorated the HFD-reduced insulin sensitivity and improved the HFD-reduced ability to regulate blood glucose levels in obese mice. The OGTT and ITT areas under the curves (AUC) were observably reduced in the HFD-ECL and HFD-ECH groups compared to the HFD group (Figure 1F,G, *p* > 0.05). As shown in Figure 1H–K, the long-term consumption of HFD disrupted lipid metabolism in obese mice. Specifically, serum TC, TG, and LDL-C levels in the HFD group were significantly higher than those in the ND group, and HDL-C levels were lower (Figure 1K, *p* < 0.05). However, administration of 10^8^ CFU/mL *E. cristatum* significantly decreased the TC and LDL-C concentrations induced by HFD (*p* < 0.05).

### 3.2. E. cristatum Promoted Thermogenesis in HFD-Fed Mice

We further evaluated the effects of *E. cristatum* on the thermogenic ability of adipose tissues in obese mice. A 12-week intake of HFD significantly increased WAT, including iWAT, rWAT, and eWAT, and the liver weight of mice (*p* < 0.05), whereas 10^8^ CFU/mL *E. cristatum* significantly decreased the weight of iWAT and rWAT in HFD-fed mice (Figure 2A, *p* < 0.05). The intervention with *E. cristatum* significantly mitigated the increase in the iWAT index caused by HFD (Figure 2B,C, *p* < 0.05), while there were no significant changes observed in the iBAT index following the intervention with *E. cristatum* (*p* > 0.05). The H&E staining further confirmed that administration of *E. cristatum* normalized the presence of multiple small lipid droplets of iBAT in obese mice, and the arrangement and uniformity of cells in WAT was improved (Figure 2D). 

As depicted in Figure 2E, in HFD-fed mice, thermogenic and mitochondrial-related genes were expressed at lower levels than in ND mice, while 10^8^ CFU/mL *E. cristatum* significantly increased the expression of Ucp1, Prdm16, and Nrf1 by 1.91-, 3.30-, and 1.69-fold (*p* < 0.05). The effects of *E. cristatum* on Ucp1 expression were further confirmed using immunofluorescence staining, in which the UCP1 and PRDM16 protein levels in mouse iBAT and iWAT exhibited evident increases upon *E. cristatum* administration (Figure 2F,G).

### 3.3. E. cristatum Facilitated the Cecal SCFAs in HFD-Fed Mice 

HFD feeding for 12 weeks caused a significant decrease (*p* < 0.05) in the content of mouse cecal SCFAs, which were dose-dependently upregulated following treatment with *E. cristatum* (Figure 3A, *p* < 0.05). In particular, in contrast to ND, HFD significantly reduced the cecal contents of propionate and acetate by 42.19% and 70.67%, respectively (*p* < 0.05), while treatment with 10^8^ CFU/mL *E. cristatum* significantly increased their levels by 41.59% and 140.55% in obese mice, respectively (Figure 3B, *p* < 0.05).

### 3.4. Impact of E. cristatum on Gut Microbiota Composition in HFD-Fed Obese Mice 

The gut microbiota can influence the host’s energy balance and metabolic processes [23,33]. Sobs and Chao indexes revealed no significant statistical differences among the groups assessed for alpha-diversity (Figure 4A,B). In Figure 4C,D, the beta-diversity analysis, which includes principal component analysis (PCoA) and non-metric multidimensional scaling (NMDS), is shown. These analyses indicated there were clear differences between the groups fed ND and HFD in terms of microbiota composition. Hierarchical clustering analysis revealed that the gut microbial communities from the groups receiving HFD or *E. cristatum* treatment could not be distinctly separated, indicating a lack of clear differentiation between these groups (Figure 4E). Via the characterization of predominant taxa at the phylum level, our analysis revealed that the consumption of HFD significantly reduced *Bacteroidetes* abundance (*p* < 0.05) while simultaneously increasing Firmicutes abundance, while the administration of 10^8^ CFU/mL *E. cristatum* mitigated these HFD-induced abnormal changes in obese mice (Figure 5A,B). Following this, genus-level clustering heatmaps of the gut microbiota were drawn based on relative classification results (Figure 5C). Specifically, intervention with 10^8^ CFU/mL *E. cristatum* significantly reduced the relative abundances of *Anaerotruncus* and *Lachnoclostridium* in obese mice induced by HFD (Figure 5D,E, *p* < 0.05), and also significantly increased the HFD-diminished relative abundances of *Parabacteroides* and *A. muciniphila* (Figure 5F,G, *p* < 0.05).

To clarify the associations of *E. cristatum*-altered gut microbiota with the obesity, fat thermogenesis, and lipid parameters, we next performed PLS correlation analysis (Figure 6). The *Firmicute* abundance correlated negatively with the IBAT index (r = −0.916, *p* < 0.05) and positively with TC levels (r = 0.947, *p* < 0.05). *Bacteroidetes* was significantly correlated with both body weight (r = −0.818, *p* < 0.01) and the iWAT index (r = −0.958, *p* < 0.05). Of note, at the genus level, *A. muciniphila* and *Faecalibaculum* exhibited negative correlations with body weight (r = −0.801, *p* < 0.05; r = −0.854, *p* < 0.05). Meanwhile, a significant negative relationship was identified between *A. muciniphila* and the iWAT index (r = −0.977, *p* < 0.01).

### 3.5. Gut A. muciniphila Mediated E. cristatum-Induced Adipocyte Thermogenesis

The weight change trends of mice in each group during the 12 weeks are shown in Figure 7A,B. The body weights of all groups were initially similar, while the mice in the HFD group exhibited a noticeable increase in body weight after 12 weeks of intervention (*p* < 0.05). Conversely, the administration of gut *A. muciniphila* significantly reduced the body weight of mice on an HFD (*p* < 0.05). Among obese mice, there was no significant difference in food intake as shown in Figure 7C. This finding suggests that the anti-obesity effect observed in the *A. muciniphila* treatment group was not attributable to the regulation of food intake. The results of the OGTT and ITT revealed that HFD-fed mice showed impaired glucose metabolism and insulin resistance, whereas *A. muciniphila* administration improved insulin sensitivity and exerted a better tolerance to glucose load in HFD mice (Figure 7D,E). Moreover, 12 weeks of HFD feeding caused lipid metabolic disorders, as evidenced by significant increases in serum TC, TG, and LDL-C and significant reductions in serum HDL-C, and *A. muciniphila* treatment remarkably normalized these parameters of lipid metabolic abnormality (Figure 7F–I).

As depicted in Figure 8A, *A. muciniphila* intervention for a continuous 12 weeks significantly decreased the HFD-augmented weight of iWAT and eWAT in mice. This was supported by H&E staining-based histopathological examination (Figure 8B), revealing that *A. muciniphila*-treated mice had much smaller adipocyte sizes than HFD mice in iWAT and eWAT. qRT-PCR assays showed that treatment with *A. muciniphila* effectively upregulated the relative expressions of Pgc1α, Prdm16, and Ucp1 in iWAT by 1.80-fold, 1.84-fold, and 2.20-fold, respectively (*p* < 0.05, Figure 8C). The immunofluorescent staining in Figure 8D confirms the enhanced expression of Ucp1 and Prdm16 in mouse iWAT following treatment with *A. muciniphila*. Additionally, in the iBAT of obese mice, *A. muciniphila* treatment significantly increased the relative mRNA levels of Pgc1α, Prdm16, and Ucp1 by 2.61-fold, 1.71-fold, and 2.07-fold (Figure 8E, *p* < 0.05). Furthermore, the administration of *A. muciniphila* also resulted in increased protein levels of UCP1 and PRDM16 in the mouse iBAT (Figure 8F).

## 4. Discussion

Fu brick tea (FBT), as a type of post-fermented tea, has been favored by consumers with an improvement in health awareness [34]. The growth of *E. cristatum*-dominated microorganisms during the last storage fermentation is an essential manufacturing process, where *E. cristatum,* as the “golden flower” fungus, is the main controller of the formation of flavor and nutritional quality of FBT [35,36]. In this study, for the first time, we found that *E. cristatum* increased energy expenditure in HFD-fed mice, reflected by the dose-dependent decrease in *E. cristatum* in the body weight of obese mice with no significant difference in food intake. Furthermore, 12 weeks of continuous administration of *E. cristatum* resulted in significant improvements in lipid and glucose disorders associated with HFD intake. Hence, the *E. cristatum* inhibitory effects on body weight, dyslipidemia, and dysglycemia are likely due to its ability to increase energy expenditure by inducing the activation of iBAT or browning of iWAT. Furthermore, the gut microbiota has been confirmed to be a critical endogenous factor in enduring effects on iBAT activation and browning of iWAT [37]. Herein, supplementation with *E. cristatum* effectively reversed HFD-induced *Firmicutes* growth and *Bacteroidetes* decline in HFD-fed mice, in which both of these phyla were validated to be associated with obesity, suggesting that the gut microbiota plays an important role in regulating the increase in energy expenditure caused by *E. cristatum* [38,39]. It is noteworthy that *A. muciniphila*, a species of mucin-degrading bacteria, has been linked to obesity [28]. In this study, administration of *E. cristatum* resulted in a substantial augmentation in the abundance of *A. muciniphila*, consequently leading to a concomitant decrease in fat accumulation. Based on these data, it would be worthwhile to verify the vital character of *A. muciniphila* in enhancing energy expenditure facilitated by *E. cristatum* consumption, and further explore the molecular mechanism underlying *E. cristatum*-induced adipocyte thermogenesis.

The existing literature demonstrates that activation of a thermogenic program by enhancing iBAT activity or inducing the browning of iWAT presents a promising avenue for increasing energy expenditure against obesity [40,41]. Adipocyte size enlargement occurs during excessive energy storage, and adipocytes become hypertrophic [40]. Our results showed that the consumption of HFD led to an increased weight of WAT and increased adipocyte size, while *E. cristatum* reversed these changes in obese mice. Furthermore, at the molecular level, treatment with *E. cristatum* increased the UCP1 and PRDM16 expressions in mouse iBAT and iWAT (Figure 2). Thus, our data make the important point that dietary *E. cristatum* treatment can effectively prevent fat accumulation in adipose tissues by increasing iBAT activity and promoting iWAT browning.

It is well established that the gut microbiota affects glucose and lipid metabolism in the host, and a comprehensive study has emphasized the importance of the gut microbiota in promoting iWAT browning [42,43]. SCFAs, primary metabolites derived from the gut microbiome and a source of energy for colonic epithelial cells, have been confirmed to be associated with the prevention and amelioration of obesity [33]. This study demonstrated a significant increase in colonic SCFA production, particularly propionic acid and acetic acid, after *E. cristatum* administration. Propionate has been demonstrated to stimulate the secretion of leptin in adipocytes via G protein-coupled receptors. Additionally, the gut microbiota-induced leptin-AMPK/STAT3 pathway has been implicated in the remodeling of beige adipocytes [44]. Another study found that acetic acid was significantly correlated with the weight of BAT [45]. Taken together, SCFAs may contribute to *E. cristatum*-induced adipocyte thermogenesis in mice.

As a critical prebiotic fungus in FBT, *E. cristatum* exhibited significant potential in restoring gut microbiota disorders and exerting anti-obesity effects [8]. Previous studies have indicated that several edible fungi can enhance energy expenditure by increasing lipid β-oxidation and promoting glucose metabolism [46,47]. However, there was no existing literature that described the fat-thermogenic effects of *E. cristatum*. In our previous work, we demonstrated that FBT mediated thermogenic function and energy expenditure via remodeling of the gut microbiota [48,49]. Hence, the functional exploration of whether the gut microbiota mediates the thermogenic properties of *E. cristatum* is an innovative and noteworthy approach in the realm of future obesity treatments. Interestingly, here, we found that *E. cristatum* increased the complexity of the bacterial diversity and counteracted the decline in *Bacteroidetes* while decreasing *Firmicutes* abundance. In addition, *Firmicutes* had a negative relationship with the iBAT index (r = −0.916, *p* < 0.05) and a positive relationship with TC (r = 0.947, *p* < 0.05). Conversely, *Bacteroidetes* were strongly correlated with body weight (r = −0.818, *p* < 0.01) and the iWAT index (r = −0.958, *p* < 0.05). These correlations suggest that the thermogenic effects induced by *E. cristatum* in adipose tissues may be mediated by the gut microbiota. Consistent with our findings, the remodeling effect of green propolis on WAT was associated with an increase in *Firmicutes* and a decrease in *Bacteroidetes* [50]. An increase in gut bacterial diversity and a decrease in the *Firmicutes*/*Bacteroidetes* ratio were associated with panax-notoginseng saponin-induced thermogenesis in obese mice [51]. Another important finding from the present study is that *E. cristatum* significantly increased the genera *Parabacteroides* and *A. muciniphila*, and reduced *Anaerotruncus* and *Lachnoclostridium* in obese mice. Notably, *A. muciniphila* exhibited a negative correlation with body weight (r = −0.854, *p* < 0.05) and a strong positive correlation with the iWAT index (r = −0.977, *p* < 0.01). Therefore, based on our findings, it can be inferred that the species *A. muciniphila* may enhance the effectiveness of *E. cristatum* in treating obesity by promoting thermogenic mobilization and reducing lipid accumulation.

Interestingly, gut *A. muciniphila,* as a promising probiotic which can be selectively proliferated by *E. cristatum* ingestion, is closely associated with host health and plays a crucial role in obesity [52,53]. In our study, we further investigated whether *A. muciniphila* could effectively improve obesity by enhancing the thermogenic mobilization of *E. cristatum*. As a result, *A. muciniphila* supplementation led to significant reductions in body weight and improvements in lipid and glucose metabolism in obese mice, indicating that *A. muciniphila* had the ability to increase the energy expenditure of obese mice (Figure 7). The process of fat thermogenesis, which involves increasing BAT activation and promoting WAT browning, is widely acknowledged as a mechanism for energy expenditure [20,46]. As expected, *A. muciniphila* significantly decreased the weight of iWAT, harbored smaller brown adipocytes, and reduced the size of iWAT, and treatment with *A. muciniphila* also resulted in an increase in the expression of UCP1 and PRDM16 in the mouse iWAT, both at the mRNA and protein levels. This suggests that *A. muciniphila* can induce iWAT browning. It was also found that *A. muciniphila* treatment significantly increased the expression of the thermogenic genes Pgc1, Prdm16, and Ucp1 in the iBAT of HFD-fed mice, and immunofluorescent staining further showed that *A. muciniphila* increased the expression of UCP1 and PRDM16. Accumulating evidence supports the finding that individuals with obesity exhibit a significant decline in the abundance of *A. muciniphila* [27,54]. Consequently, the administration of *A. muciniphila* effectively alleviated obesity [27]. However, despite the positive impact of increased *A. muciniphila* levels on obesity, there are still restrictions that prevent its direct administration or use as a dietary supplement [55]. Our findings might support the hypothesis that *E. cristatum* exerts a gut microbiota-dependent effect for improving obesity by enhancing the relative abundance of *A. muciniphila*. *A. muciniphila* produces acetic acid [56], and *E. cristatum* significantly enhances the acetic acid concentration in obese mice. In turn, acetic acid promotes the relative abundance of *A. muciniphila*. *E. cristatum*, *A. muciniphila*, and SCFAs contribute to therapeutic strategies for obesity and obesity-related metabolic disorders. It is worth noting that *A. muciniphila* is not the exclusive source of acetates [57]; therefore, we will further explore the mechanisms of *E. cristatum* influence on the relative abundance of *A. muciniphila* using pseudo-germ-free mice or fecal microbiota transplantation in the future.

## 5. Conclusions

*E. cristatum*, the dominant fungal probiotic in FBT, suppressed obesity and ameliorated obesity-induced lipid accumulation and hyperglycemia by playing a critical role in upregulating the thermogenesis of adipose tissues. *E. cristatum* also ameliorated abnormalities in the gut microbiome and its metabolites, in which it especially enriched colonic *A. muciniphila* and acetic acid. Consistently, supplementation with *A. muciniphila* further revealed its involvement in *E. cristatum*-induced adipocyte thermogenesis. Overall, this study provided evidence for the compelling interactions between *E. cristatum*, *A. muciniphila*, acetic acid, and thermogenesis, paving the way for the confirmation of *E. cristatum* as an edible anti-obesity prebiotic fungus and *A. muciniphila* as a dietary supplement in the near future.

## Figures and Tables

**Figure 1 foods-12-03716-f001:**
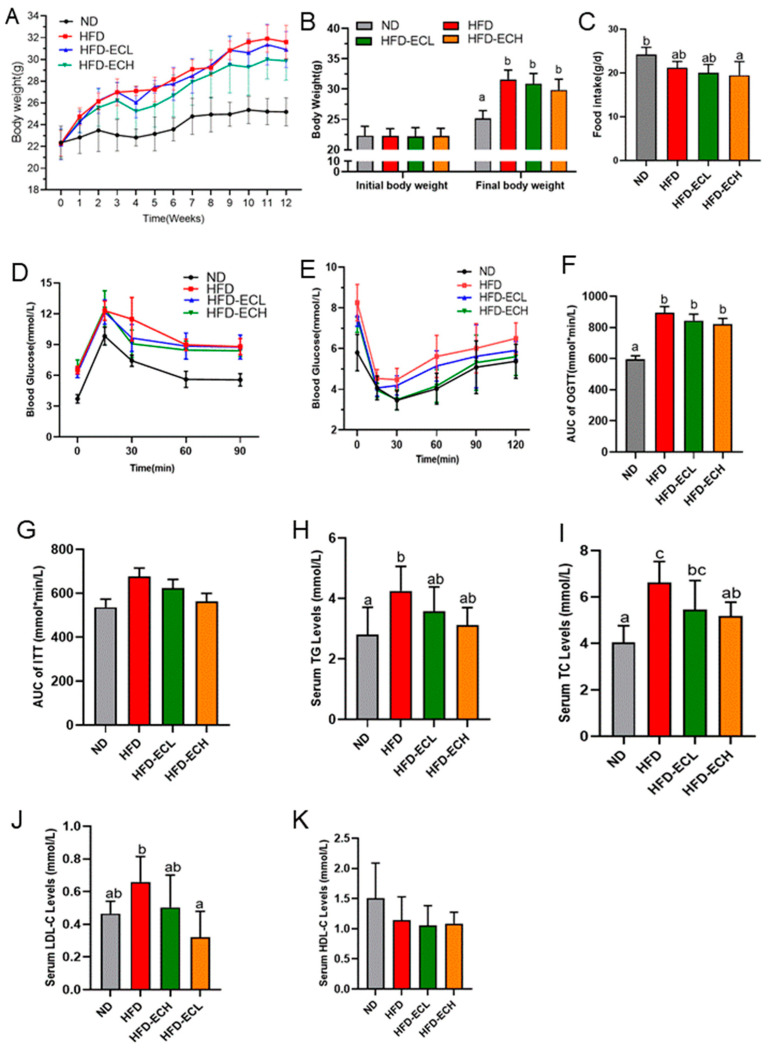
Curves of body weight (**A**). The gain in body weight (**B**). Food intake (**C**). OGTT (**D**). ITT (**E**). AUC of OGTT, ITT (**F**,**G**). Levels of TG, TC, LDL-C, and HDL-C (**H**–**K**). Different letters are used to denote statistical significance (*p* < 0.05) among groups.

**Figure 2 foods-12-03716-f002:**
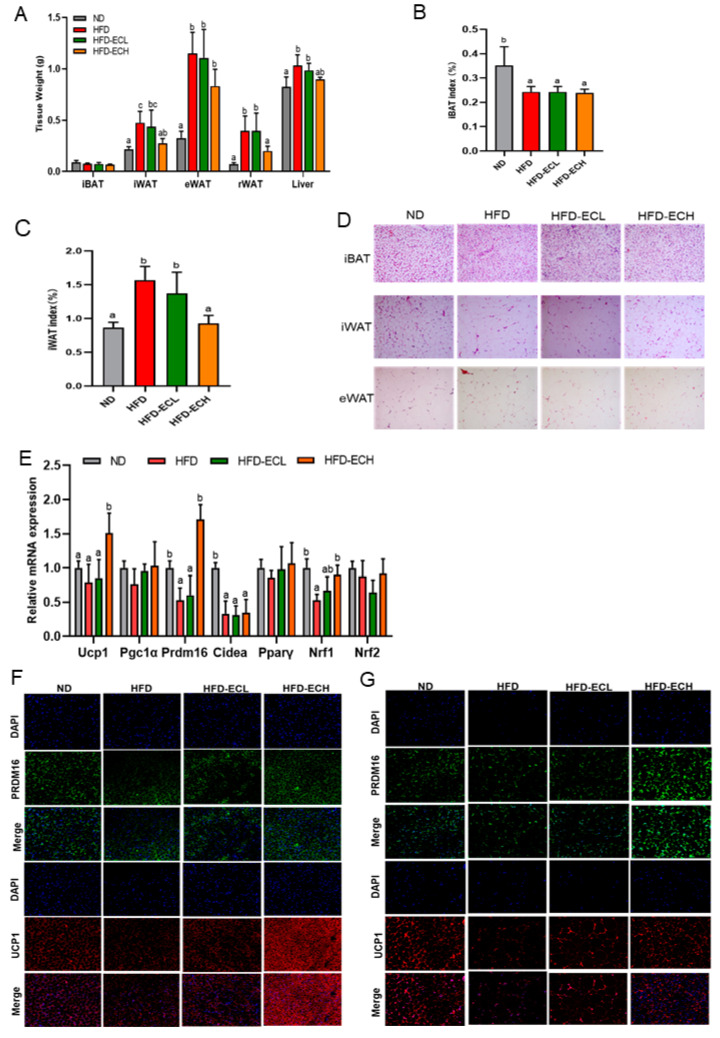
The weight of iBAT, rWAT, eWAT, iWAT, and the liver (**A**). Tissue index of iBAT and iWAT (**B**,**C**). H&E staining of iWAT, iBAT, and eWAT (**D**). Thermogenic and mitochondrial gene expression in iBAT (**E**). PRDM16 and UCP1 immunofluorescent staining in iBAT (**F**) and iWAT (**G**). Adipocyte histology was observed at a magnification of 20×. Different letters are used to denote statistical significance (*p* < 0.05) among groups.

**Figure 3 foods-12-03716-f003:**
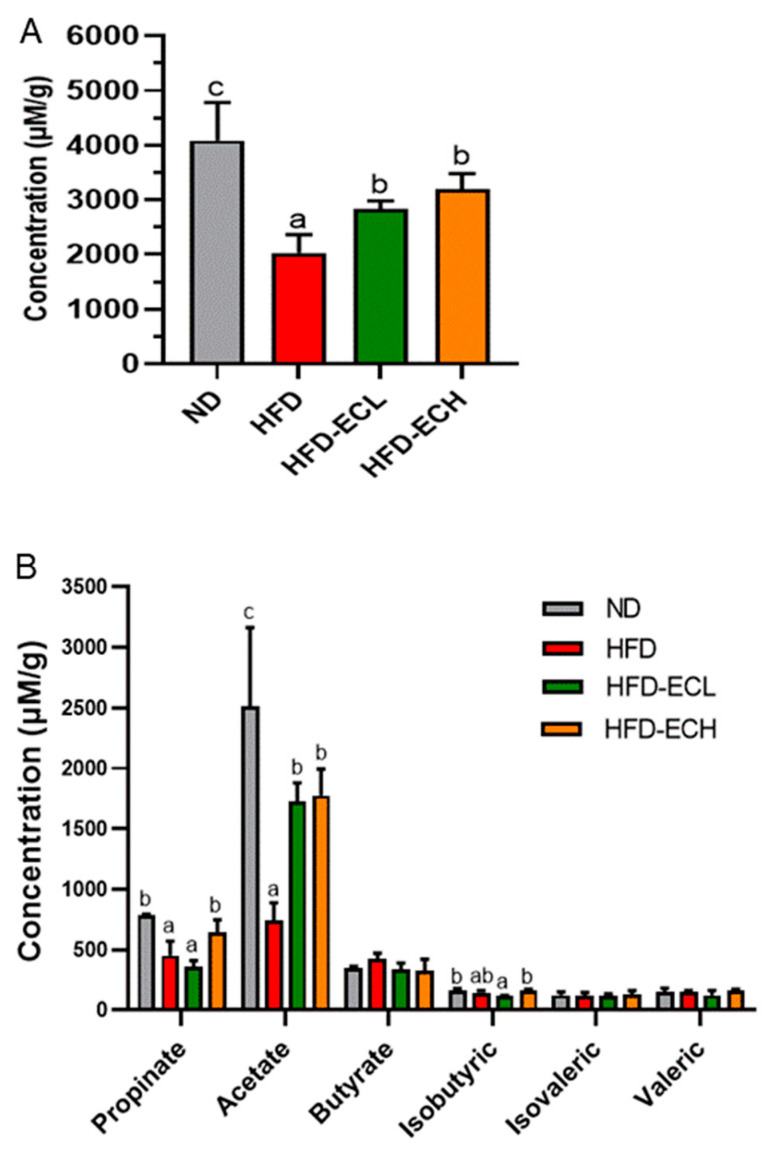
The total concentration of SCFAs (**A**) and the concentration of 6 different kinds of SCFA (**B**). Different letters are used to denote statistical significance (*p* < 0.05) among groups.

**Figure 4 foods-12-03716-f004:**
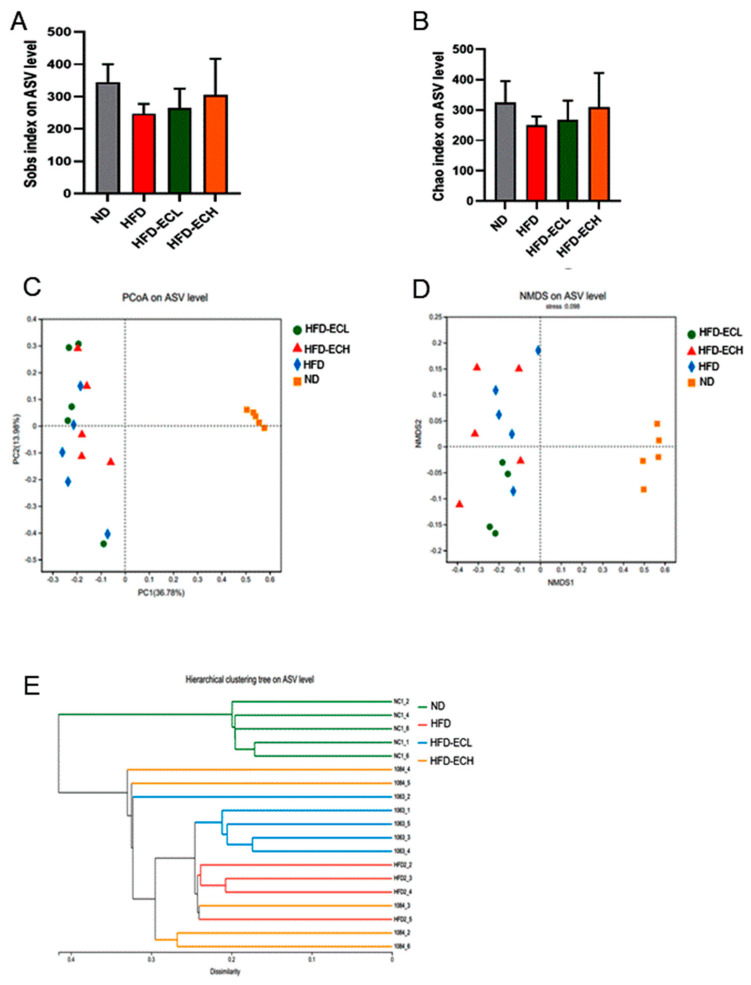
Alpha-diversity including Sobs index, Chao index, PCoA, and NMDS (**A**–**D**), and hierarchical clustering (**E**) of microbiota communities.

**Figure 5 foods-12-03716-f005:**
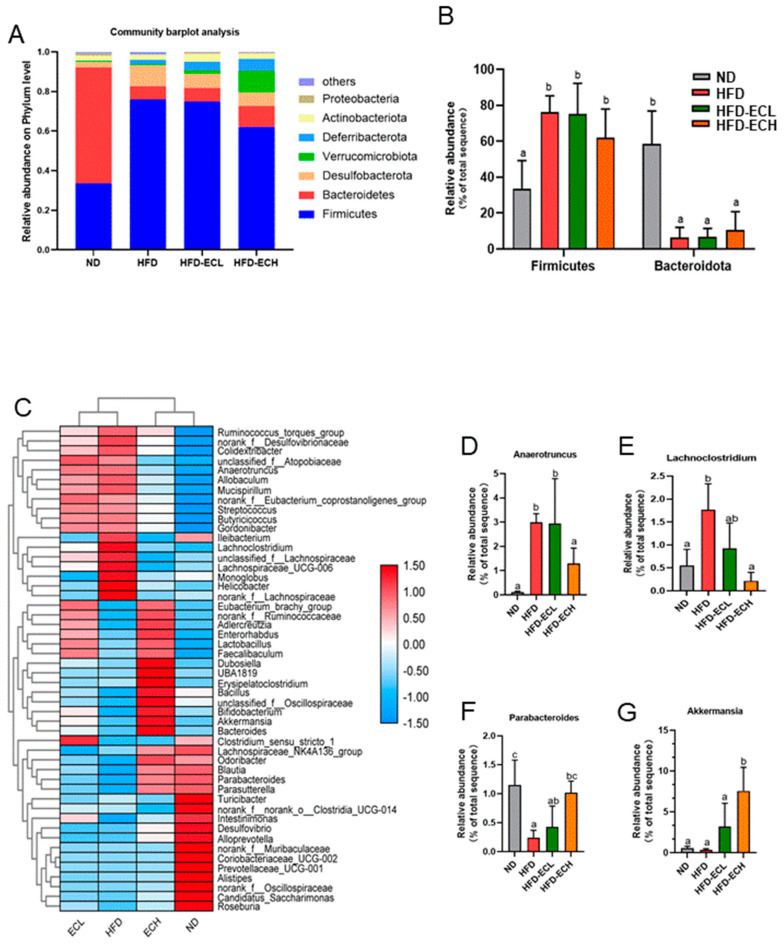
Phylum-level analysis of gut microbiota (**A**). Relative abundances of *Bacteroidetes* and *Firmicutes* (**B**). Genus-level heatmap of gut microbiota (**C**). Relative abundances of *Anaerotruncus*, *Lachnoclostridium*, *Parabacteroides*, and *A. muciniphila* (**D**–**G**). Different letters are used to denote statistical significance (*p* < 0.05) among groups.

**Figure 6 foods-12-03716-f006:**
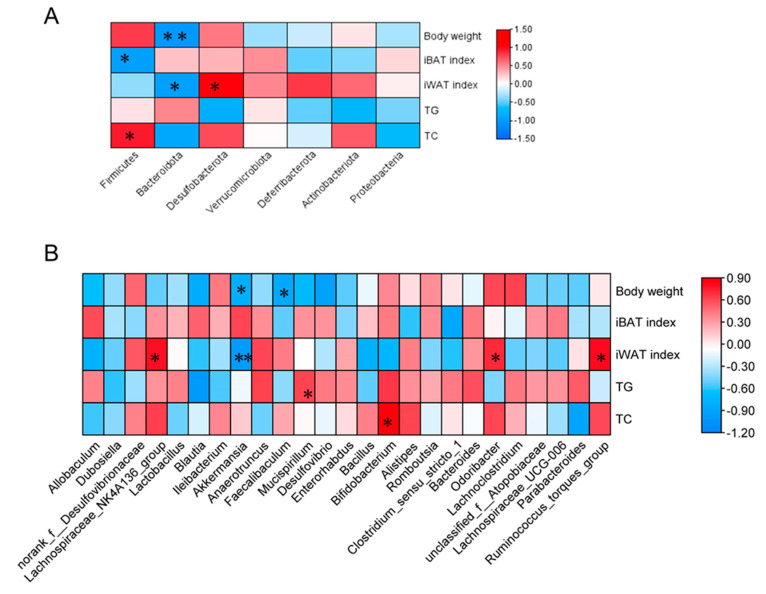
The relationship between obesity-related parameters and gut microbiota at phylum and genus levels (**A**,**B**). * *p* < 0.05, ** *p* < 0.01.

**Figure 7 foods-12-03716-f007:**
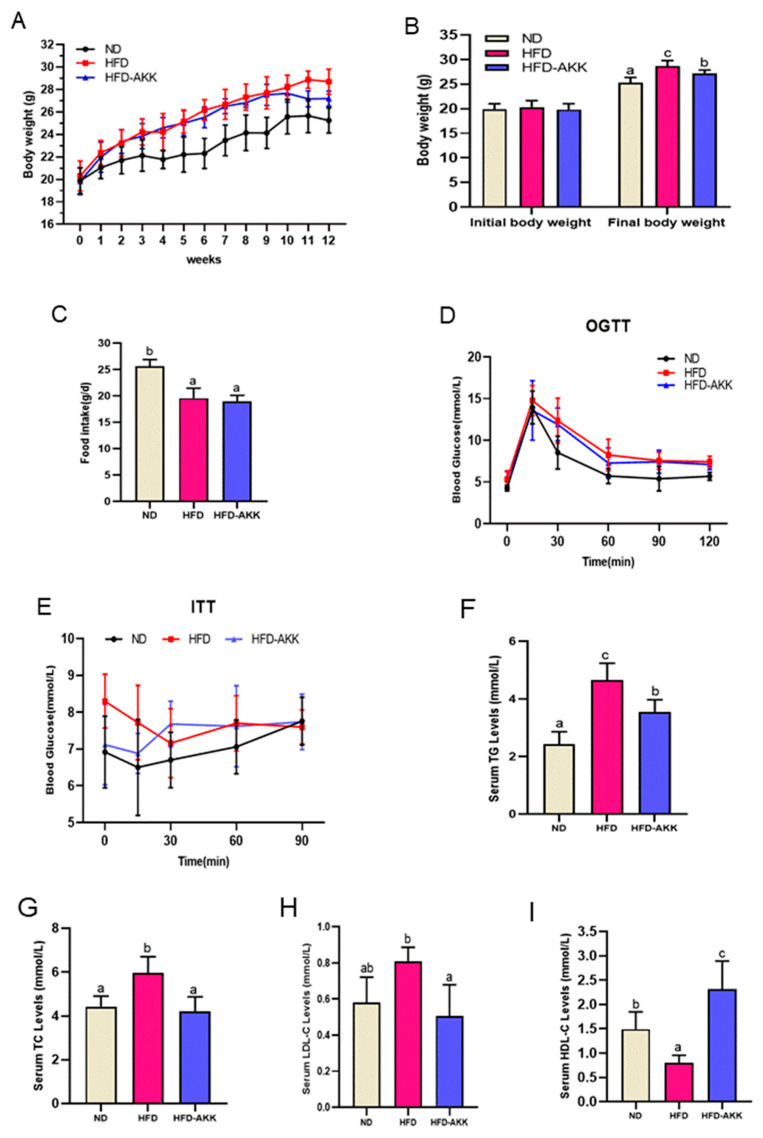
Curves of body weight (**A**). The gain in body weight (**B**), Food intake (**C**), OGTT and ITT (**D**,**E**). Levels of TG, TC, LDL-C, and HDL-C (**F**–**I**). Different letters are used to denote statistical significance (*p* < 0.05) among groups.

**Figure 8 foods-12-03716-f008:**
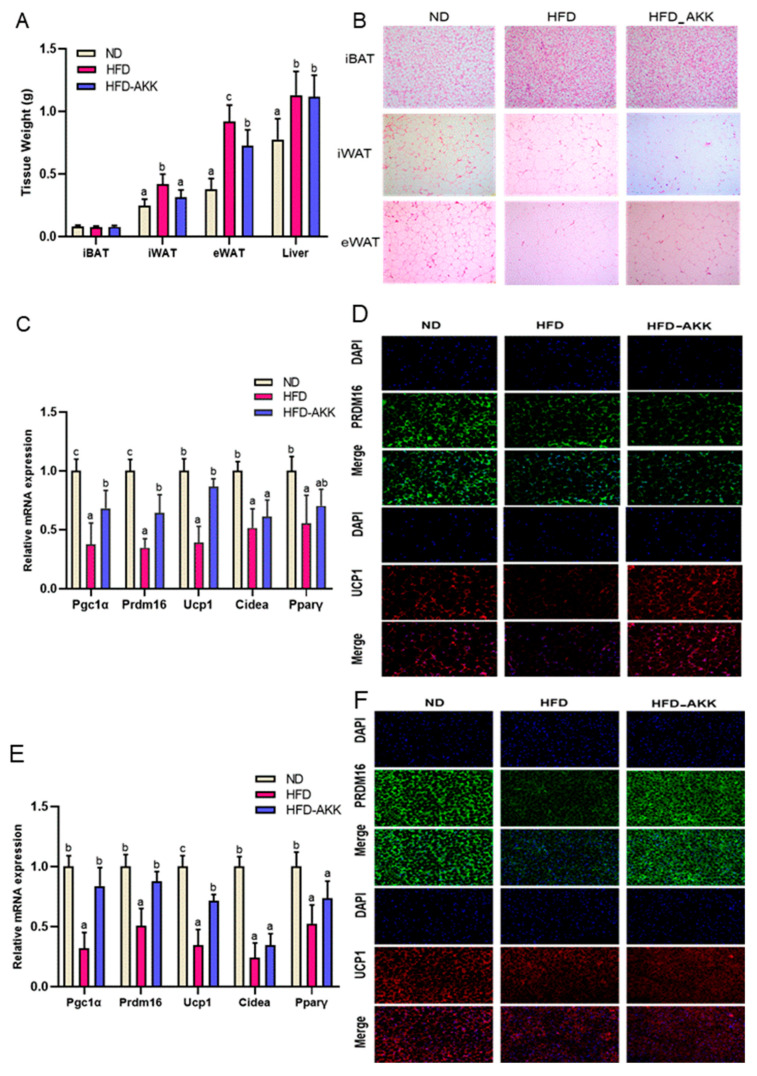
The weight of iBAT, rWAT, eWAT, iWAT, and the liver (**A**). H&E staining of iWAT, iBAT, and eWAT (**B**). Thermogenic and mitochondrial gene expression in iWAT (**C**). PRDM16 and UCP1 immunofluorescent staining in iWAT (**D**). Thermogenic and mitochondrial gene expression in iBAT (**E**). PRDM16 and UCP1 immunofluorescent staining in iBAT (**F**). Adipocyte histology was observed at a magnification of 20×. Different letters are used to denote significance (*p* < 0.05) among groups.

**Table 1 foods-12-03716-t001:** Primers used for qRT-PCR.

Gene	Forward Primer	Reverse Primer
Pgc1α	GTGTGTGCTGTGTGTCAGAG	AACCAGAGCAGCACACTCTAT
Prdm16	CATGTGCGAAGGTGTCCAAA	GTCACCGTCACTTTTGGCT
Pparγ	GACGCGGAAGAAGAGACCTG	TCACCGCTTCTTTCAAATCTTGT
Ucp1	GTGAACCCGACAACTTCCGA	TGGCCTTCACCTTGGATCTGA
Cidea	AGGCCGTGTTAAGGAATCTGC	AACCAGVVTTTGGTGCTAGG
Nrf1	CTGCAGGTCCTGTGGGAAT	GGCTCTGAGTTTCCGAAGCA
Nrf2	AGGCCGTGTTAAGGAATCTGC	TATCCAGGGCAAGCGACTCA

## Data Availability

The data used to support the findings of this study can be made available by the corresponding author upon request.

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
