# Peer review of "Eurotium cristatum from Fu Brick Tea Promotes Adipose Thermogenesis by Boosting Colonic Akkermansia muciniphila in High-Fat-Fed Obese Mice"

_foods, 2023, doi:10.3390/foods12203716_

Round 1

Reviewer 1 Report

The presented article entitled ‘Eurotium cristatum from Fu-Brick Tea Promotes Adipose Thermogenesis by Boosting Colonic Akkermansia muciniphila in High Fat-Fed Obese Mice’ The study appears to be comprehensive and uses molecular, microbiological, and histological techniques.

However, as with any study, there are areas for potential improvement:

Plagiarism: 43 % reduce it up to a permissible limit

Abstract:

The abstract is quite lengthy. Consider condensing the information while retaining key findings and their implications. Specify the dosage of E. cristatum used for the different treatments. "Dose-dependent" is mentioned, but the specifics are not clear. The correlation coefficients can be moved to the main text to make the abstract more concise. Avoid redundancy; for instance, mentioning the enhancement of thermogenic activity by both E. cristatum and A. muciniphila can be streamlined.

Introduction:

Introduce the main topic (E. cristatum and its potential effects) earlier to give readers an immediate sense of the study's focus. Consider emphasizing the significance of E. cristatum and its relationship with FBT more prominently. When mentioning prior studies, it may be beneficial to briefly state their findings to provide context. Transition more smoothly between discussing adipose tissues and the gut microbiota to make the progression of topics more logical. When first introducing A. muciniphila, briefly explain its significance in relation to the study's focus.

Methodology:

The method used for the administration of E. cristatum (e.g., mixed with food, injected, etc.) should be explicitly mentioned. Specify the total number of mice used and the exact number in each group. Clearly state the duration of each phase of the experiment. Detail any observational criteria or measurements taken during the experiments, beyond weight and food consumption. Ensure a consistent control method for both experiments to make results more comparable. More details should be provided for techniques like GC-MS and 16S rRNA sequencing. For GC-MS: What are the conditions (temperature, column type, etc.)? For 16S rRNA sequencing: Specify the primers used and the database for taxonomic assignments. Clarify the criteria for statistical significance and specify which tests are used for which comparisons. Some statements, like the conditions of the animal experiments, seem to be repeated. Ensure concise representation of methods.

Results:

In some sections, the results are quite dense, making them slightly difficult to comprehend. Simplifying and clarifying results, especially for a general audience, would be helpful. While many of the figures and results are discussed, the inclusion of more explicit statements about the significance and implications of the findings could strengthen this section. The extensive use of figures is good, but ensuring clear and comprehensible labeling, legends, and descriptions is crucial for interpretation. Any anomalies or unexpected findings in the figures should be discussed. It is essential to provide more explicit details about the statistical analyses used. While p-values are mentioned, mentioning the actual test (e.g., ANOVA, t-test) used for each analysis could provide better clarity. Ensure that the necessary corrections for multiple comparisons were made where appropriate, as this can influence the validity of the findings.

Discussion:

Comparison with previous research is vital. It would be helpful to compare the results more extensively with the existing literature to validate findings and place them in a broader context. While the study does delve into some mechanisms (e.g., the role of A. muciniphila), more in-depth discussion on potential underlying mechanisms would enrich the study. For instance, how might E. cristatum influence A. muciniphila populations? Every study has limitations. It would be beneficial to address the limitations of this study explicitly, whether they be related to the mouse model used, potential confounding factors, or other issues. Discuss the potential broader implications and future research directions based on the findings. How might these results influence future obesity treatments or the development of functional foods?

Conclusions:

The conclusion summarizes the findings well. However, emphasizing the novelty or significance of these findings in the broader field of obesity research could add value. It would be beneficial to include recommendations for future research or potential applications of these findings in real-world scenarios. Ensure that the conclusion is written in a way that's accessible to a wider audience, as this section is often what many readers (especially non-experts) will refer to understand the main takeaways.

In summary, addressing potential limitations, and clearly laying out implications and recommendations would further enhance the paper's quality and impact.

Minor editing of English language required

Reviewer 2 Report

The authors of the manuscript "Eurotium cristatum from Fu-Brick Tea Promotes Adipose Thermogenesis by Boosting Colonic Akkermanisa muciniphila in Hight Fat-Fed Obese Mice" examined the effect of E. cristatum on thermogenesis and the inhibitory effect on obesity in mice fed a diet containing 30% lard. The work contains interesting research elements that still require confirmation in model studies on the human factor. The manuscript is well edited and prepared. Research supported by properly selected statistical analyses. The only reservations related to the work concern chapter 2, where few additions are required. In section 2.1, please provide how E. cristatum was isolated from Fu-Brick tea and what different concentrations of E. cristatum mushroom suspension concentrates were prepared. In section 2.2, specify whether the oral doses introduced were the same throughout the experiment? .Section 2.4 specify exactly which test sets were used to determine the biochemical parameters of mouse blood.

Round 2

Reviewer 1 Report

Accept in Present form 

Author Response

We deeply appreciate the reviewer's insightful comments on our manuscript. As the reviewer's comments and suggestions for authors in Round 2 indicated that our manuscript is deemed acceptable in its present form, we will follow your recommendation and maintain the current version.